# Effect of Carbonation Treatment on the Strength and CO_2_ Uptake Rate of Composite Cementitious Material with a High Steel Slag Powder Content

**DOI:** 10.3390/ma16186204

**Published:** 2023-09-14

**Authors:** Zhimin He, Xuyang Shao, Xin Chen

**Affiliations:** 1Department of Civil Engineering, Ningbo University, Ningbo 315211, China; hezhimin@nbu.edu.cn (Z.H.); 13136315116@163.com (X.C.); 2Collaborative Innovation Center of Coastal Urban Rail Transit, Ningbo University, Ningbo 315211, China

**Keywords:** carbonation, steel slag powder, metakaolin, strength, CO_2_ uptake

## Abstract

As a major steel producer, China is now eager to develop feasible solutions to recycle and reuse steel slag. However, due to the relatively poor hydration activity of steel slag, the quantity of steel slag used as a supplemental binder material is limited. In order to improve the cementitious properties of steel slag, the strength and carbonation degree of the high-content steel slag powder–cement–metakaolin composite cementitious material system under CO_2_ curing conditions were investigated. The compressive strengths of the mortar specimens were tested and compared. The carbonation areas were identified and evaluated. A microscopic analysis was conducted using X-ray diffraction (XRD), thermogravimetry analysis (TG), and scanning electron microscopy (SEM) to reveal the chemical mechanisms. The results showed that CO_2_ curing significantly increased the early strength as the 3D compressive strength of the specimens increased by 47.2% after CO_2_ curing. The strength of the specimens increased with increasing amounts of metakaolin in a low water-to-binder ratio mixture. The 3D compressive strength of the specimens prepared with 15% metakaolin at a 0.2 water-to-binder ratio achieved 44.2 MPa after CO_2_ curing. Increasing the water-to-binder ratio from 0.2 to 0.5 and the metakaolin incorporation from 0% to 15% resulted in a 25.33% and 19.9% increase in the carbonation area, respectively. The calcium carbonate crystals that formed during carbonation filled the pores and reduced the porosity, thereby enhancing the strength of the mortar specimens. The soundness of the specimens after CO_2_ curing was qualified. The results obtained in the present study provide new insight for the improvement of the hydration reactivity and cementitious properties of steel slag powder.

## 1. Introduction

With the rapid development of the steel industry, China has become the largest steel producer worldwide. Steel slag is a by-product produced during the steel-making process, accounting for approximately 15–20% of the total steel production [1,2]. In 2021, steel slag production in China reached 120 million tons with a cumulative stockpile of over 800 million tons [3]. However, the comprehensive utilization rate is only 30% [4,5]. Mostly, steel slag is disposed of at landfills as solid waste, occupying land resources and causing environmental issues, since steel slag contains a variety of heavy metals which can contaminate soil and underground water after leaching [3,6]. The steel industry is eager to develop practical, economical, and environmentally friendly approaches to reuse steel slag.

Steel slag contains potential cementitious components such as dicalcium silicate (C_2_S), tricalcium silicate (C_3_S), tetracalcium aluminoferrite (C_4_AF), and other minerals [7,8]. However, a large number of highly hydrated active minerals in steel slag decompose or transform during the production procedures due to the high temperatures (1600–1700 °C) and slow cooling process [9]. Moreover, the presence of non-reactive components, such as the divalent metal oxide solid solution (RO phases), further reduces the hydration activity of steel slag [10]. In addition, free calcium oxide (*f*-CaO) and free magnesium oxide (*f*-MgO) in steel slag continue to hydrate after cement hardening and cause volume expansion, expansion, and cracking damages, which were often noticed when utilizing steel slag as construction materials [11]. Therefore, proper treatments have to be conducted in order to improve the usage of steel slag in construction applications.

On the other hand, steel slag is highly carbonation active owing to the presence of *f*-CaO, *f*-MgO, C_3_S, and C_2_S, which would react with CO_2_ and form carbonates [1,12,13]. Compared to standard curing, carbonization curing can effectively improve the strength and durability of steel slag powder specimens [14,15,16,17,18,19]. Zhang et al. [14] found that carbonization can improve the early strength of the specimens and that carbonation improved the compressive strength of ladle steel slag and electric arc furnace steel slag binders by 48.1 MPa and 25 MPa, respectively, compared to conventional hydration, reaching a 12-h paste strength of 52.8 MPa and 27 MPa, respectively. Park and Chang et al. [15,16] found that carbonation curing can effectively reduce the drying shrinkage and water absorption of steel slag concrete and improve its resistance to sulfate attacks. Polettini et al. [20] found that the carbonization reaction of steel slag products was closely related to the CO_2_ concentration, carbonization pressure, and carbonization time. They also noted that increasing the CO_2_ concentration and carbonization pressure can increase the rate of diffusion and dissolution of CO_2_ in the specimens as well as increase the early carbonation reaction rate of the specimens. Mo et al. [1,21,22,23,24] used CO_2_ gas (≥99.9%) to cure steel slag products and reported that abundant Ca_x_Mg_1-x_CO_3_ was formed after carbonation, which contributed to the porosity reduction and strength enhancement of the specimens. Meanwhile, the soundness of the steel slag was improved due to the consumption of *f*-CaO and *f*-MgO during carbonation. Chen et al. [25] reported that steel slag absorbed CO_2_ equivalent to 12% of its own mass through a 1 h carbonation with 99.9 vol% CO_2_, indicating that the utilization of steel slag for CO_2_ sequestration as a building material was a feasible approach towards carbon neutralization. At present, the content of steel slag powder used as cementitious material is generally low. However, in a high CO_2_ concentration, a protective film of CaCO_3_ will form on the surface of the specimens, hindering internal carbonization [20,26]. Increasing the content of steel slag powder and the carbonization effect of the specimens was the focus of current research.

Metakaolin is a metastable amorphous silicoaluminate compound with fine particles and pozzolanic activity with filling effects. Currently, steel slag–metakaolin systems are mostly found in geopolymers [27,28,29]. Furlani et al. [27] found that geopolymers containing 40 wt% steel slag powder and 60 wt% metakaolin had the best strength. The Ca released from steel slag powder could facilitate the formation of C-A-S-H and C-S-H gels, which could fill the voids and bond the adjacent solid phase together, increasing the strength of the geopolymer [28]. Huang et al. [30] investigated the effect of metakaolin on a steel slag powder–cement-based cementitious system under steam curing conditions. The results showed that metakaolin greatly improved the strength of the steel slag powder–cement composite cementitious system. Metakaolin improved the carbonation of cementitious materials [31,32]. However, limited research was conducted on the correlation between the strength and carbonation degree of the high-content steel slag powder–cement–metakaolin composite material system under CO_2_ curing conditions.

In the current literature, the content of steel slag powder is generally low, and the utilization of steel slag is limited. This study aims to solve the problem of low-strength high-content steel slag specimens by adding metakaolin to improve the strength and carbonation effect of the specimens, heighten the usage of steel slag in cement-based cementitious materials, and therefore, increase the recycling ratio of steel slag. The strength, CO_2_ sequestration, and soundness of the mortar specimens were tested and compared. The pore structures, carbonation products, carbonation area, and morphology were analyzed.

## 2. Materials and Methods

### 2.1. Raw Materials

A P·O 42.5 ordinary Portland cement (PC) was provided by Ningbo Hailuo Cement Co., Ltd., Ningbo, Zhejiang, China, with a specific surface area of 365 m^2^/kg, a density of 3.1 g/cm^3^, and a loss on ignition value of 2.2%. The steel slag powder (SS) was provided by Hebei Jingye Steel Co., Ltd., Shijiazhuang, Heibei, China. It was a yellowish-brown powder with an average particle size of 74 μm (Figure 1a). The metakaolin (MK) was provided by Hunan Chaopai Metakaolin Co., Ltd., Changsha, Hunan, China. It was a white powder with an average particle size of 2 μm and a specific surface area of approximately 25,000 m^2^/kg (Figure 1b). The mineral compositions of the steel slag powder are shown in Figure 2. The chemical compositions of Portland cement, steel slag powder, and metakaolin are shown in Table 1.

The sand was an ISO standard sand (GB/T17671-2021 [33]). The water-reducing agent that was used was a polycarboxylate superplasticizer with a water-reducing rate of ≥20%. Tap water was used as the mixing water. The dosage of the water-reducing agent was adjusted during mixing to ensure the consistent flowability of the different mixtures.

### 2.2. Sample Preparation

The mortar specimens with a dimension of 40 × 40 × 160 mm were prepared according to the mix proportions listed in Table 2. The steel slag powder content was fixed at 50 wt% with the cement and metakaolin in different proportions. The cement content was set at 50%, 45%, 40%, and 35% and the metakaolin content was set at 0%, 5%, 10%, and 15% accordingly. The water-to-binder ratio was set at 0.2, 0.3, 0.4, and 0.5. The sand-to-binder ratio was fixed at 2. In total, 16 different mix proportions were prepared.

Firstly, the mixtures were cast and cured in a climate chamber with a relatively humidity of 60 ± 5% and a temperature of 20 ± 2 °C for 24 h. Secondly, the mortar specimens were demolded and placed in a drying oven at a temperature of 30 °C for 24 h to remove the excess moisture. Finally, the specimens were cured in a carbonation chamber with a relatively humidity of 70 ± 5%, a temperature of 20 ± 2 °C, and a CO_2_ concentration of 90 ± 2% for 24 h. Tests were conducted on the specimens both before and after carbonation. The word C in MK0-2-C denotes after carbonation.

### 2.3. Test Method

#### 2.3.1. Compressive and Flexural Strength

The flexural and compressive strength tests were conducted according to the GB/T 17671-2021 [33] by using a universal testing machine. The flexural strength loading rate was set at 50 N/s and the compressive strength loading rate was set at 2.4 kN/s. Three identical specimens were prepared and the test results were averaged. The strength development rate after CO_2_ curing was calculated using Equation (1).
(1)$$r=\frac{\sigma_{c}-\sigma_{0}}{\sigma_{0}}\times 100\%$$
where *r* is the strength development rate after CO_2_ curing (%); $\sigma_{0}$ is the 2D strength of the specimen before CO_2_ curing (MPa); and $\sigma_{c}$ is the 3D strength of the specimen after CO_2_ curing (MPa).

#### 2.3.2. Carbonated Area

The carbonated area was indicated by spraying a 1 wt% of phenolphthalein alcoholic solution evenly upon the cross-section of the specimen. The carbonated area was identified by coloring a range on the cross-section after 30 s. The phenolphthalein staining image was processed and analyzed using the image analysis software Image J 1.8.0 to calculate the ratio of the carbonated area according to Equation (2). The area recognition process is shown in Figure 3.
(2)$$\mathrm{Percentage}\;\mathrm{of}\;\mathrm{carbonized}\;\mathrm{area}=\frac{\mathrm{Area}\;\mathrm{of}\;\mathrm{carbonized}\;\mathrm{area}}{\mathrm{Total}\;\mathrm{cross}-\mathrm{sectional}\;\mathrm{area}}$$

#### 2.3.3. Soundness

The specimens with water-to-binder ratios of 0.3 were selected for soundness testing. The soundness of the specimens before and after CO_2_ curing was tested using the Le Chatelier soundness test according to the GB/T1346-2011 [34] (Figure 4). The volume expansion of the specimens in the Le Chatelier test after boiling increased the opening distance of the Le Chatelier. The soundness of the specimens could be characterized according to the opening distance of the Le Chatelier.

#### 2.3.4. Pore Structures

The pore structure was tested using the evaporable water content method [35]. The specimens with water-to-binder ratios of 0.3 and 0.4 were selected for testing. The total porosity ($P_{0}$) and the porosity ($P_{1}$) of large pores (>30 nm) were calculated using Equations (3) and (4). The difference between these two values was the porosity ($P_{2}$) of small pores (≤30 nm), given by Equation (5).
(3)$$P_{0}=\frac{\rho_{c}\left(M_{0}-M_{2}\right)}{M_{0}\rho_{W}}\times 100\%$$
(4)$$P_{1}=\frac{\rho_{c}\left(M_{0}-M_{1}\right)}{M_{0}\rho_{W}}\times 100\%$$
(5)$$P_{2}=P_{0}-P_{1}$$
where $\rho_{c}$ is the density of the specimen (g·cm^3^); $\rho_{W}$ is the density of water (g·cm^3^); $M_{0}$ is the mass of the specimen after 48 h of vacuum water saturation; $M_{1}$ is the mass of the specimen after drying in a desiccator for 28 days to constant weight; and $M_{2}$ is the mass of the specimen dried at 105 °C for 48 h to constant weight.

#### 2.3.5. Carbonation Products

The samples were collected at a depth of 0~5 mm in the surface area of the specimen. The samples were ground to pass through a 200 mesh standard sieve. The carbonated product content was tested using a thermogravimetric analysis (TG-DTG) at a temperature range from 25 °C to 1000 °C with a heating rate of 10 °C/min under nitrogen atmosphere protection.

#### 2.3.6. Microstructure

The samples described in Section 2.3.5 were firstly put into anhydrous ethanol to stop hydration and dried in a vacuum drying oven at 60 °C for 48 h before testing. The microscopic morphology was observed using a scanning electron microscope (SEM-EDS) device(COXEM Co., Ltd., Seoul, Republic of Korea).

## 3. Results and Discussion

### 3.1. Compressive and Flexural Strength

The flexural strength, compressive strength, and strength development rate of the specimens are presented in Figure 5. It was noticed that the strength of all the specimens improved after 24 h of carbonation. The greatest enhancement in the compressive strength was 47.2%. It was due to the formation of CaCO_3_ after carbonation, which filled the pores and created a more compact microstructure. Additionally, CaCO_3_ had a higher microhardness compared to mineral clinker, which also enhanced the strength of the specimens [36,37,38].

Figure 5 shows that the strength development rate after carbonation was directly proportional to the amount of metakaolin incorporation under the same water-to-binder ratio. When the water-to-binder ratio was 0.5, increasing the metakaolin content from 0% to 15% resulted in an increase in the compressive strength development rate from 32.3% to 47.2%. Since the active components in metakaolin reacted with the Ca(OH)_2_ in the hydration products, they generated more C-S-H, which increased the dissolution of Ca^2+^ and benefited the carbonation reaction of the system [39,40]. Moreover, the pore-filling effect of the small particles of metakaolin was also beneficial to the system [30]. This filling effect was more noticeable at low water-to-binder ratio conditions. When the water-to-binder ratio was 0.2, increasing the metakaolin content from 0% to 15% resulted in an increase in the compressive strength from 39.2 MPa to 44.2 MPa. However, the strength of the specimens was reduced at high water-to-binder ratios due to the decreased hydration products.

As shown in Figure 5, the strength development rate after carbonation was directly proportional to the water-to-binder ratio under the same metakaolin content conditions. When the metakaolin content was 10%, increasing the water-to-binder ratio from 0.2 to 0.5 resulted in an increase in the compressive strength development rate from 15.3% to 40.3%. The reason was that the porosity of the system increased with an increase in the water-to-binder ratio and, therefore, facilitated the diffusion of CO_2_ within the mortar and improved the carbonation efficiency [41,42,43].

Figure 6 shows the comparison between the strengths of the specimens with a 10% metakaolin content under CO_2_ curing and standard curing (cured in a relatively humidity of 95 ± 5% and a temperature of 20 ± 2 °C for 48 h after being demolded). The flexural and compressive strength of the specimens after carbonation were equal to or higher than those of the standard-cured specimens when the water-to-binder ratio was 0.2, 0.3, and 0.4, respectively. However, both the flexural and compressive strength of the specimens after carbonation were slightly lower than those of the standard-cured specimens when the water-to-binder ratio was 0.5. Carbonation converted the hydration product into CaCO_3_, which filled the pores in the specimen to improve the strength. At a high water-to-cement ratio, the filling effect of carbonation was no longer obvious due to a higher number of pores in the specimen, while the reduction in the hydration product led to a reduction in the strength of the specimen [44].

### 3.2. Carbonated Area and CO_2_ Uptake

#### 3.2.1. Carbonized Area

Table 3 shows the carbonated area of the mortar specimens after 24 h of carbonation. It was clear that the carbonated area of the specimen was effectively increased with the incorporation of metakaolin. The carbonated area of the specimen was more than 50% at a 15 wt% metakaolin addition. When the water-to-binder ratio was 0.2, increasing the amount of metakaolin from 0% to 15% resulted in a 19.9% increase in the carbonated area. The pozzolanic reaction of metakaolin produced more C-S-H, which increased the dissolution of Ca^2+^ and promoted the carbonation reaction [39,40].

Since a higher water-to-binder ratio led to a more porous structure, which contributed to the diffusion of CO_2_ gas, the carbonated area of the specimens increased with an increase in the water-to-binder ratio. The carbonated area with a 0.5 water-to-binder ratio ranged from over 60% to 88.57%. Increasing the water-to-binder ratio from 0.2 to 0.5 led to a 25.3% increase in the carbonated area when no metakaolin was incorporated. When the amount of metakaolin was 15%, increasing the water-to-binder ratio from 0.2 to 0.5 led to a 56% increase in the carbonated area. The size of the carbonated area was correlated with the strength development of the specimens after carbonation. It was noticed that the specimens with larger carbonated areas exhibited higher strength development rates, indicating that carbonation contributed to the increase in strength.

#### 3.2.2. CO_2_ Uptake

The CO_2_ uptake was obtained thought a thermogravimetric analysis. The TG curves of the specimens with different metakaolin contents are shown in Figure 7. Since the decomposition temperature of the calcium carbonate was between 550 °C and 1000 °C, the CO_2_ uptake was calculated using the mass loss, according to Equation (6) [45].
CO_2_ uptake (%) = (m_550_ − m_1000_)/m_c_(6)
where m_550_ is the mass of the sample at 550 °C, m_1000_ is the mass of the sample at 1000 °C, and m_c_ is the mass of the cementitious material that was used.

According to Figure 7, the incorporation of metakaolin significantly improved the CO_2_ uptake of the specimens as the CO_2_ uptakes of MK0-3-C, MK0-5-C, MK10-3-C, and MK10-5-C were 4.8%, 9.2%, 15.2%, and 21.4%, respectively. The CO_2_ uptakes of the specimens with a 10% metakaolin content exhibited a 217% and 133% increase, respectively, compared to the specimens without metakaolin. The specimens with a 0.5 water-to-binder ratio increased the CO_2_ uptake by 92% and 41% compared to the specimens with a 0.3 water-to-binder ratio, suggesting that the increase in the water-to-binder ratio had a positive effect on the carbonation of the specimens.

### 3.3. The Soundness Evaluation

Soundness is the key technical index for steel slag powder used in construction materials and is a decisive factor in the building industry. Table 4 lists the soundness of the specimens before and after CO_2_ curing. According to GB/T20491-2017 [46], steel slag powder specimens should not exceed 5 mm after the boiling of the Le Chatelier opening distance. According to Table 4, the Le Chatelier opening distances were more than 7 mm before CO_2_ curing. After 24 h of CO_2_ curing, the soundness of the specimens was significantly improved. On the other hand, the opening distance gradually reduced with an increase in metakaolin. For example, the opening distance of the MK0-3-C specimen was 3.5 mm and that of the MK15-3-C specimen was only 2.5 mm. The soundness of all the specimens met the specification requirements. Usually, the volume expansion caused by the hydration of *f*-CaO and *f*-MgO in steel slag powder is the main reason for the poor soundness of steel slag powder. However, when *f*-CaO and *f*-MgO became stable CaCO_3_ or Ca_x_Mg_1-x_CO_3_ after CO_2_ curing [23], they effectively improved the soundness of the steel slag powder specimens. The positive effect of metakaolin on the carbonation also enhanced the soundness of the specimens after CO_2_ curing.

### 3.4. Porosity Profiles

The 28-d porosity of the mortar specimens with water-to-binder ratios of 0.3 and 0.4 were tested under standard and CO_2_ curing conditions using the evaporable water content method, and the results are shown in Table 5. Compared to standard curing, the total and macropore volume of the mortar significantly decreased after CO_2_ curing. The total and macropore volumes of the MK0-3 specimen decreased by 9.4% and 54.2%, respectively, after CO_2_ curing. This was owing to the pores that were effectively filled by nano-calcium carbonate during carbonation, which reduced the total and macropore porosity of the specimens. This also explained the strength enhancement of the specimens after CO_2_ curing, as discussed previously.

Since the particle size of metakaolin was smaller than that of the steel slag powder and cement, the pore-filling effect further improved the quality of the mortar specimens. Moreover, the pozzolanic reaction of metakaolin provided C-S-H filling in the large pores, which also reduced the porosity and increased the density. The porosity of the specimens after CO_2_ curing firstly decreased and then increased with the increase in the metakaolin content. The total porosity of MK10-3-C decreased by 45.3%, 28.5%, and 7.8%, compared to MK0-3-C, MK5-3-C, and MK15-3-C, respectively. The effect of the cement reduction resulted in reduced hydration products, leading to an increased porosity [47]. The increased metakaolin content compensated for the effect of the cement reduction due to the increased hydration products and pore-filling effect of the particles.

### 3.5. Carbonation Products

The TGA analysis was conducted on the specimens with four mixtures (MK0-3, MK10-3, MK0-5, and MK10-5) to analyze the changes in the phase content during carbon curing. The TG-DTG curves are shown in Figure 8. The DTG curves suggested that the changes in the loss on ignition before and after CO_2_ curing were concentrated in the decomposition of C-S-H, Ca(OH)_2_, and CaCO_3_ at 50~200 °C, 400~450 °C, and 550~1000 °C, respectively.

All the samples without CO_2_ curing showed a significant mass loss area between 400 to 450 °C, indicating a high Ca(OH)_2_ content before CO_2_ curing. The mass loss of each sample decreased significantly between 400 to 450 °C after CO_2_ curing, while the mass loss increased significantly between 550 to 1000 °C, indicating that a significant amount of Ca(OH)_2_ was carbonated during the CO_2_ curing process. The mass loss according to the decomposition of CaCO_3_ in the MK0-3-C, MK10-3-C, MK0-5-C, and MK10-5-C samples was 3.7%, 6.1%, 11.7%, and 14.3%, respectively. This indicated that both the increasing the water-to-binder ratio or the metakaolin content would benefit from the carbonation reaction.

### 3.6. Microstructure

Figure 9 shows the SEM images of the specimens before and after CO_2_ curing. According to Figure 9a,b, the MK10-5 specimen before CO_2_ curing showed a loose structure with many large capillary pores between 5 and 10 μm and the presence of hexagonal platelet Ca(OH)_2_, reticular calcium silicate hydrate (C-S-H), and elongated prismatic ettringite (Aft). As shown in Figure 9c,d, Ca(OH)_2_ and Aft were no longer observed inside the MK10-5-C specimen. The calcite crystals were generated inside to fill the pores, which made a dense structure and improved the strength. The amount of carbonation products generated inside the MK10-5-C sample was significantly higher than that in the MK10-3-C sample. Figure 10 presents the ESD energy spectrum analysis of M10-3-C. It was evident that these crystalline materials were CaCO_3_. Meanwhile, it was noticed that Mg reacted with CaCO_3_ to form Ca_x_Mg_1−x_CO_3_, indicating that MgO leached and precipitated with the calcium carbonate in the carbonation reaction.

## 4. Conclusions

The strength, CO_2_ sequestration, and soundness of the high-content steel slag powder–cement–metakaolin composite material system under CO_2_ curing conditions were tested and compared. The pore structures, carbonation products, carbonation area, and morphology were analyzed. Some conclusions can be drawn as follows.

The incorporation of metakaolin effectively improved the strength development rate of the steel slag powder composite cementitious material after carbonation. With a metakaolin content of 15%, the compressive strength development rate after carbonation reached a maximum of 47.2%. At low water-to-binder ratio conditions, the strength of the slag cement mortar after carbonation increased with an increase in the metakaolin content. Carbonation also enhanced the early strength of the specimens. For the specimens with 15% metakaolin content and a water-to-binder ratio of 0.2, the 3-day strength after carbonation achieved 44.2 MPa. Carbonation improved the early strength of the specimens, providing a basis for the use of steel slag powder specimens as bricks, blocks, etc. The development of further strength will continue to be investigated.The cementitious system of steel slag powder–cement–paraffin showed an excellent CO_2_ uptake performance. The CO_2_ uptake of the specimen with a 10% metakaolin content at a 0.5 water-to-binder ratio reached 21.4%. The CO_2_ uptake and carbonized area of the specimen was increased by increasing the water-to-binder ratio or metakaolin content. The carbonated area of the specimen with a 0.5 water-to-binder ratio and a 15% metakaolin content achieved 88.57%, indicating that the utilization of steel slag for CO_2_ sequestration as a building material is a feasible approach towards carbon neutralization.CO_2_ curing managed to reduce the content of *f*-CaO and *f*-MgO in the steel slag powder and improve the soundness of the steel slag powder cement mortar. The Le Chatelier opening distance of the carbonated steel slag powder specimens were smaller than 3.5 mm. Moreover, the incorporation of metakaolin also enhanced the soundness of the slag cement mortar. However, the durability of the carbonization specimens deserves a subsequent study.The formation of calcium carbonate crystals during CO_2_ curing filled the internal pores of the specimens. The pozzolanic activity and filling effect of metakaolin also contributed to the reduction in the porosity. As a result, the total porosity and macropore volume significantly decreased. Therefore, the strength of the specimens was enhanced. The lowest porosity was achieved at a 10% metakaolin content.

## Figures and Tables

**Figure 1 materials-16-06204-f001:**
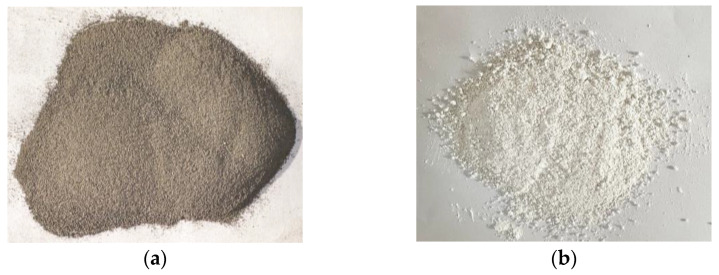
Steel slag powder (**a**) and metakaolin (**b**).

**Figure 2 materials-16-06204-f002:**
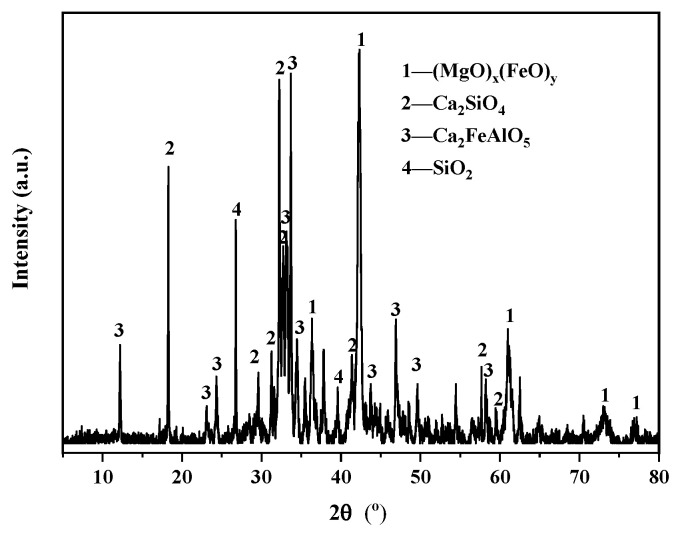
XRD pattern of the steel slag.

**Figure 3 materials-16-06204-f003:**
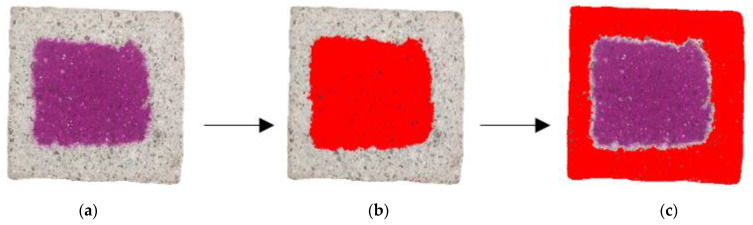
Carbonated area identification and calculation: (**a**) phenolphthalein staining image; (**b**) uncarbonated area; (**c**) carbonated area.

**Figure 4 materials-16-06204-f004:**
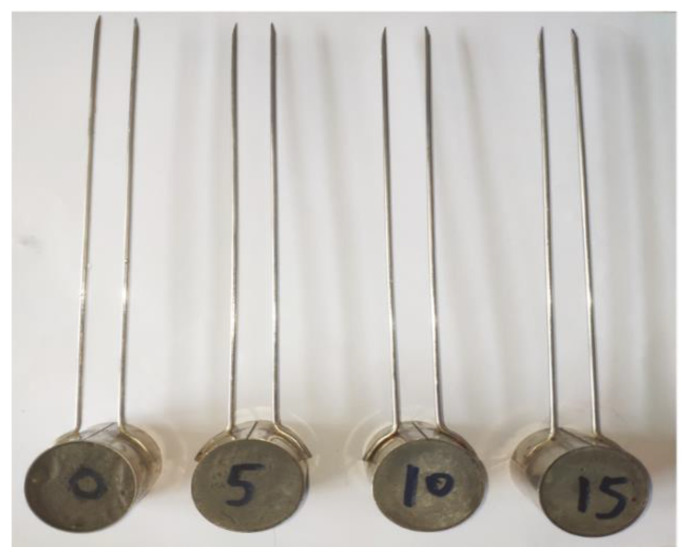
Le Chatelier soundness test.

**Figure 5 materials-16-06204-f005:**
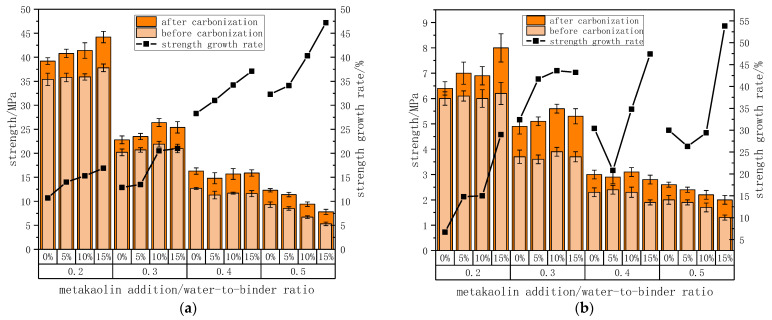
Strength of the specimens before and after carbonation with different metakaolin additions and water-to-binder ratios: (**a**) compressive strength; (**b**) flexural strength.

**Figure 6 materials-16-06204-f006:**
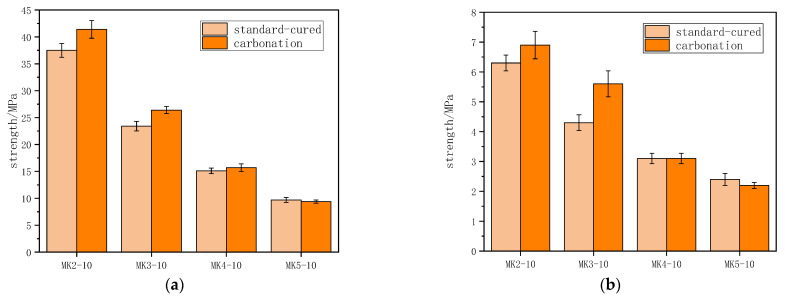
Strength of the specimens of standard-cured and carbonation with different water-to-binder ratios: (**a**) compressive strength; (**b**) flexural strength.

**Figure 7 materials-16-06204-f007:**
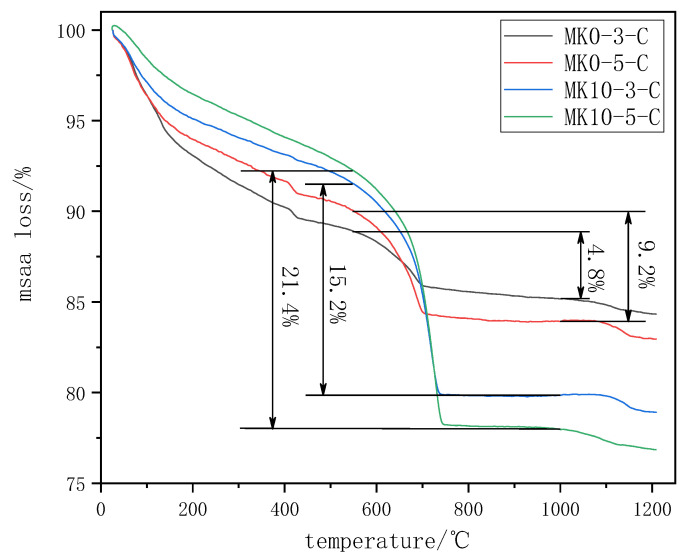
TG curves of the specimens after carbonation.

**Figure 8 materials-16-06204-f008:**
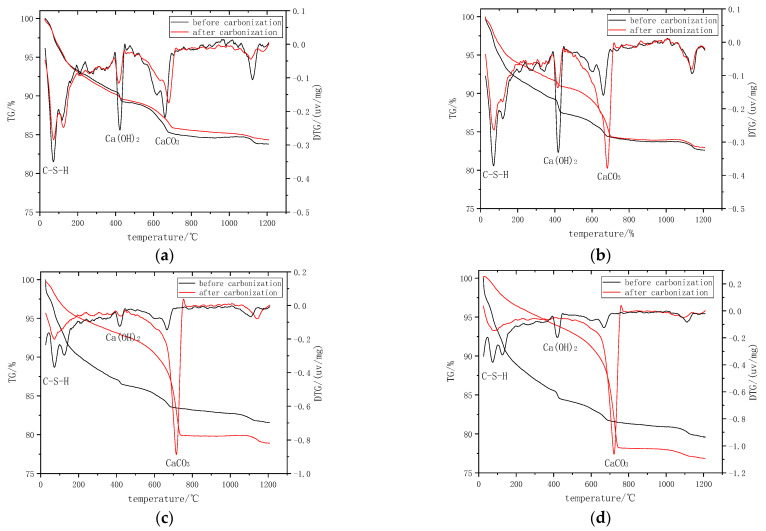
TG-DTG of the specimens before and after carbonation: (**a**) MK0-3, (**b**) MK10-3, (**c**) MK0-5, and (**d**) MK10-5.

**Figure 9 materials-16-06204-f009:**
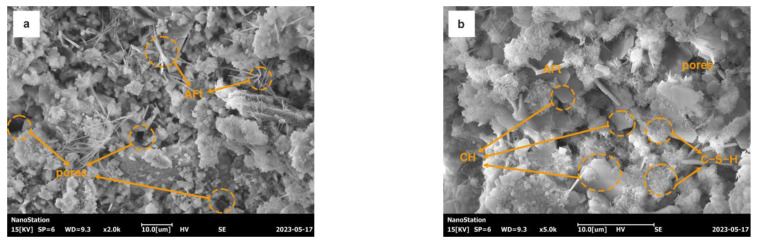
SEM of the specimens before and after carbonation: (**a**,**b**) MK10-5 before carbonation; (**c**,**d**) MK10-5-C; and (**e**,**f**) MK10-3-C.

**Figure 10 materials-16-06204-f010:**
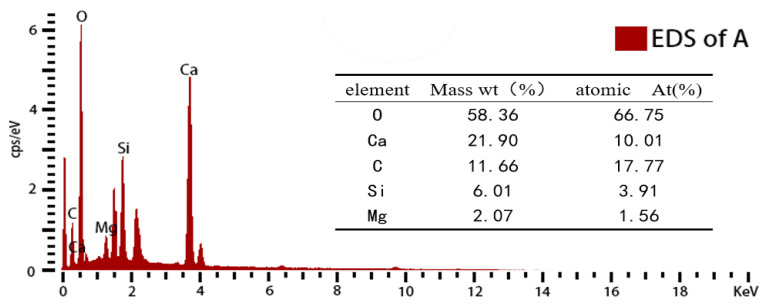
EDS of the specimens after carbonation.

**Table 1 materials-16-06204-t001:** Chemical compositions of the Portland cement, steel slag powder, and metakaolin (wt%).

	CaO	SiO_2_	Fe_2_O_3_	Al_2_O_3_	MgO	SO_3_
PC	55.2	24.5	3.5	4.5	4.0	2.0
SS	42.5	13.1	29.5	3.0	5.9	0.3
MK	0.3	64.8	1.0	29.1	1.6	0.2

**Table 2 materials-16-06204-t002:** Mix proportion of the mortar specimens.

Specimen	Steel Slag Powder (wt%)	Cement (wt%)	Metakaolin (wt%)	Water-to-Binder Ratio	Sand-to-Binder Ratio
MK0-2 ^1^	50	50	0	0.2/0.3/0.4/0.5	2
MK5-2		45	5		
MK10-2		40	10		
MK15-2		35	15		

^1^ MK0-2 represents 0% of the metakaolin and a 0.2 water-to-binder ratio.

**Table 3 materials-16-06204-t003:** Carbonated area of the mortar with different metakaolin contents and water-to-binder ratios.

Water-to-Binder Ratio	0.2	0.3	0.4	0.5
0% metakaolin	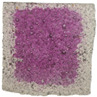	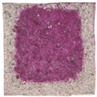	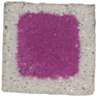	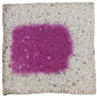
	36.87%	41.87%	49.19%	62.20%
5% metakaolin	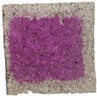	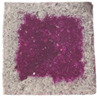	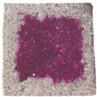	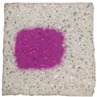
	39.72%	45.42%	57.35%	73.78%
10% metakaolin	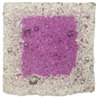	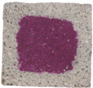	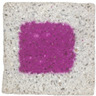	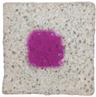
	58.25%	56.36%	63.81%	82.37%
15% metakaolin	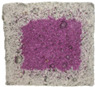	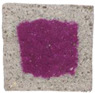	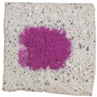	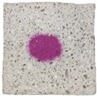
	56.77%	57.16%	72.58%	88.57%

**Table 4 materials-16-06204-t004:** Soundness test results.

Specimens	before Carbon Curing		after Carbon Curing	
	Opening Distance	Whether the Soundness Is Qualified	Opening Distance	Whether the Soundness Is Qualified
MK0-3	7.8 mm	no	3.5 mm	yes
MK5-3	7.5 mm	no	3.2 mm	yes
MK10-3	7.0 mm	no	3.1 mm	yes
MK15-3	7.1 mm	no	2.5 mm	yes

**Table 5 materials-16-06204-t005:** 28-d porosity of the mortar specimens.

	Number	Macropore (>30 nm)		Microporous (≤30 nm)		Total Porosity (%)
		Porosity (%)	Percentage	Porosity (%)	Percentage	
Standard curing	MK0-3	1.124	10.47%	9.617	89.53%	10.742
	MK5-3	0.750	9.08%	7.513	90.92%	8.263
	MK10-3	0.552	8.22%	6.163	91.78%	6.715
	MK15-3	0.528	8.11%	5.982	91.89%	6.510
	MK0-4	2.145	15.12%	12.044	84.88%	14.189
	MK5-4	1.628	14.27%	9.775	85.73%	11.403
	MK10-4	1.052	13.36%	6.824	86.64%	7.876
	MK15-4	1.177	13.87%	7.310	86.13%	8.487
CO_2_ curing	MK0-3	0.515	5.29%	9.213	94.71%	9.728
	MK5-3	0.353	4.75%	7.090	95.25%	7.443
	MK10-3	0.131	2.46%	5.188	97.54%	5.319
	MK15-3	0.233	4.04%	5.534	95.96%	5.767
	MK0-4	1.612	12.80%	10.982	87.20%	12.594
	MK5-4	1.213	11.11%	9.705	88.89%	10.918
	MK10-4	0.655	9.77%	6.045	90.23%	6.700
	MK15-4	0.744	9.94%	6.736	90.06%	7.480

## Data Availability

The data used to support the findings of this study are available from the corresponding authors upon request.

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
