# Peer review of "Effect of Carbonation Treatment on the Strength and CO2 Uptake Rate of Composite Cementitious Material with a High Steel Slag Powder Content"

_materials, 2023, doi:10.3390/ma16186204_

Round 1

Reviewer 1 Report

The manuscript addresses the topic of increasing the use of steelmaking slag in cement-based materials, important from the point of view of sustainable development.

The effect of conducting analyzes in this area is to be a potential increase in the recycling rate of steel slag.

The topic is original and justified in the indicated field.

So far, no similar variant of the analysis has been carried out in this regard.

The methodology is clearly described.

The conclusions are consistent with the evidence and arguments presented. The conclusions correspond to the main task set.

The list of references is exhaustive.

Tables and figures are clear and legible.

Author Response

Dear Reviewer:

manuscript number:materials-2553813 

We would like to thank you for your careful reading, helpful comments, and constructive suggestions, which has significantly improved the presentation of our manuscript. We have carefully considered all comments from you and revised our manuscript accordingly. In this revised version, all revisions to the manuscript are highlighted by using yellow background color.

Reviewer 2 Report

The paper “Effect of carbonation treatment on the strength and CO2 uptake rate of composite cementitious material with a high steel slag powder content” is a good paper and has been well written. However, there are some comments that should be revised to be acceptable. The comments are as following points:

1.   In the abstract, it's better to mention the problem statement first then the objective, method, results, conclusion, and recommendations.

2.   In the keywords part, the word “CO2 uptake” should end with a full stop (.).

3.   Improve English and correct grammar mistakes are necessary.

4.   At the end of the introduction part, the authors should focus on the lack of knowledge in the point that you are investigating (use of the high volume of steel slag as cement replacement). Who studied this topic? And which point did not examine yet?

5.   Other comments in the PDF file.

Author Response

Dear Reviewer :

manuscript number:materials-2553813 

We would like to thank you for your careful reading, helpful comments, and constructive suggestions, which has significantly improved the presentation of our manuscript. We have carefully considered all comments from you and revised our manuscript accordingly. In this revised version, all revisions to the manuscript are highlighted by using yellow background color:

Comment 1. In the abstract, it's better to mention the problem statement first then the objective, method, results, conclusion, and recommendations.

Response:Thank you for the above suggestions. We have added the problem statement, objective, and recommendations in the abstract. 

Comment2. In the keywords part, the word “CO2 uptake” should end with a full stop (.).

Response:Thank you so much for your careful check. We have added a full stop (.) at the end of the word “CO2 uptake”. 

Comment3. Improve English and correct grammar mistakes are necessary.

Response:We have thoroughly checked and corrected the grammatical errors and typos we found in our revised manuscript.

 Comment4.At the end of the introduction part, the authors should focus on the lack of knowledge in the point that you are investigating (use of the high volume of steel slag as cement replacement). Who studied this topic? And which point did not examine yet?

Response:Thank for your comments. We have added more related research and literature in the introduction section. In the end, we have analyzed the lack of current research on the carbonisation of steel slag products.

Reviewer 3 Report

C02-curing of cementitious materials appears as a very promising solution to help decarbonize the civil engineering industry and increase the early age performance of cementitious materials. Several recent studies have been published on the subject but the understanding of its impact on complex formulations is still needed.

This manuscript investigates the ‘Effect of carbonation treatment on the strength and CO2 uptake rate of composite cementitious material with a high steel slag powder content’. To this end, several mortar specimens are formulated with various metakaolin (MK) amount and water-to-binder ratio and various tests (mechanical, chemical) are performed.

The manuscript is divided into four sections: an introduction which is rather good but lacks references to fully highlight the novelty of the paper, a Materials and Methods section which describes the main experimental features, a good Results and discussion section and a conclusion summarizes the main observations.

Some additions are needed, especially concerning the literature review:

1)      Abstract: ‘the soundness of the specimens after was also evaluated’. After carbonation?

2)  l 44: write RO in full letters

3) l 57: the authors used a very high C02 concentration of 99%. Some sentences about the impact of CO2 concentration  are needed

4) l 60-66: some references about the synergistic role of metakaolin and slag, especially at early age on micromechanical and macromechanical properties can be added. Also metakaolin grade (high-grade, low-grade) has an impact and should be evocated.

5) l 117-199: precise the age of specimens in the strength growth rate index calculation

6) What is the objective of Le Chatelier test? Chatelier takes a capital C

7) Change example 3 and example 4 to equations 3 and 4

8) Fig 5 and 6: add error bars

9) Fig 5: if possible, remove the 3 line segments between 0.2 MK 15% and 0.3 MK0%, 0.3 MK 15% and 0.4 MK0% and 0.4 MK 15% and 0.5 MK0%,

10) l 197 define ‘standard curing

11) table 3: remove ‘carbonated area’ in the first column

12) fig 9 caption: correct sample names

13) conclusion: recap the objective of the paper at the beginning of the conclusion

14) conclusion: give some perspectives

Author Response

Dear Reviewer :

manuscript number:materials-2553813 manuscript

 We would like to thank you for your careful reading, helpful comments, and constructive suggestions, which has significantly improved the presentation of our manuscript. We have carefully considered all comments from you and revised our manuscript accordingly. In this revised version, all revisions to the manuscript are highlighted by using yellow background color:

Comment 1. Abstract: ‘the soundness of the specimens after was also evaluated’. After carbonation?

Response:We are very sorry for our negligence, this is an evaluation of soundness after carbonization. We have changed "the soundness of the specimens after was also evaluated" to "The soundness of the specimens after CO2 curing was qualified" in line 27.

Comment 2. l 44: write RO in full letters

Response:We have wrote the full letters of RO phases in line 48,"the divalent metal oxide solid solution". 

Comment3. l 57: the authors used a very high CO2 concentration of 99%. Some sentences about the impact of CO2 concentration  are needed

Response:Thank for your comments.We have added some sentences and literatures about the impact of CO2 concentration in line 63-67.

Comment4. l 60-66: some references about the synergistic role of metakaolin and slag, especially at early age on micromechanical and macromechanical properties can be added. Also metakaolin grade (high-grade, low-grade) has an impact and should be evocated.

Response:Thank you for your suggestion. Currently, steel slag-metakaolin systems are mostly found in geopolymer,so we have added some references about steel slag-metakaolingeopolymer,as well as some references about the effect of metakaolin on carbonation in line 80-92.

Comment5. l 117-199: precise the age of specimens in the strength growth rate index calculation

Response:Thank you so much for your careful check .We have precised the age of specimens in the strength growth rate index calculation in line145-147, after CO2 curing is 3d strength and before CO2 curing is 2d strength.

Comment6. What is the objective of Le Chatelier test? Chatelier takes a capital C

Response:We have added the objective of Le Chatelier test in line 165-167 and changed "Le chatelier" to "Le Chatelier" in the article.

Comment7. Change example 3 and example 4 to equations 3 and 4

Response:We have changed "example 3 and example 4" to "Eq. (3) and (4)" in line 173

Comment8. Fig 5 and 6: add error bars

Response:We gratefully appreciate for your valuable suggestion, we have added add error bars in Fig 5 and 6.

Comment9. Fig 5: if possible, remove the 3 line segments between 0.2 MK 15% and 0.3 MK0%, 0.3 MK 15% and 0.4 MK0% and 0.4 MK 15% and 0.5 MK0%,

Response:We have removed the 3 line segments between 0.2 MK 15% and 0.3 MK0%, 0.3 MK 15% and 0.4 MK0% and 0.4 MK 15% and 0.5 MK0% inFig 5.

Comment10. l 197 define ‘standard curing

Response:We have defined"standard curing" in line 220-221, "cured in a relatively humidity of 95±5%, a temperature of 20±2 °C for 48h after demoulded".

Comment11. table 3: remove ‘carbonated area’ in the first column

Response:We have remove "carbonated area" in the first column in table 3.

Comment12. fig 9 caption: correct sample names

Response:We are very sorry for our negligence,we have changed fig 9 caption in line 351.

Comment13. conclusion: recap the objective of the paper at the beginning of the conclusion

Response:We have recapped the objective of the paper at the beginning of the conclusion in line 356-359.

Comment14. conclusion: give some perspectives

Response:We have added some perspectives in conclusion section.

Round 2

Reviewer 2 Report

Accept

Reviewer 3 Report

The reviewers comments have been correctly addressed.